# A New Approach to Evaluate the Sustainability of Ecological and Economic Systems in Megacity Clusters: A Case Study of the Guangdong–Hong Kong–Macau Bay Area

Hui Li [1,2,*], Xue Huang [1], Qing Xu [1], Shuntao Wang [1], Wanqi Guo [1], Yan Liu [1], Yilin Huang [1] and Junzhi Wang [1]

1 College of Forestry and Landscape Architecture, South China Agricultural University, Guangzhou 510642, China
2 Guangdong Rural Construction Research Institute, Guangzhou 510642, China
* Correspondence: ydlihui@scau.edu.cn

**Abstract:** An emergy analysis is used to assess the sustainability of urban agglomerations' eco-economic systems, which are generally measured by emergy–value sustainability indicators using a combination of several system indicators. However, this assessment approach is not applicable to economically developed high-density urban agglomerations. The application of the traditional entropy value evaluation method needs to be expanded to further strengthen the sustainability of the complex eco-economic–social relationships in megacity cluster regions. In this study, taking the Guangdong–Hong Kong–Macau Greater Bay Area (GBA) as a case study, we study a new evaluation method for evaluating the sustainable development capacity of cities. This method is based on the entropy power method and is used to construct the evaluation system of all indicators of the social–economic–natural subsystems of the eco-economic system, and it couples the development degree with the coordination degree. (1) This study shows that the new method is applicable for the sustainability assessment of high-density megacity clusters and is more accurate and comprehensive. The sustainability rankings are provided for Zhaoqing, Jiangmen, Huizhou, Guangzhou, Macau, Foshan, Zhongshan, Dongguan, Zhuhai, and Shenzhen. Hong Kong is the most representative, with a high sustainability index, but has the lowest level of coordination and a clear incoherence within the system. (2) The current emergy structure of the GBA city cluster is extremely unreasonable. The GBA city cluster is a resource-consuming city with a common characteristic of a low level of coordinated development. Although urban clusters have some potential in terms of renewable emergy and resources, the recycling rate of waste is low, and the consumption rate of nonrenewable resources is high. The effective use of land resources has become an important factor in the bottlenecking of sustainable development, and all other cities face such problems, except Zhaoqing, Jiangmen, and Huizhou. (3) The GBA city cluster can be divided into three categories according to the new method. Category 1 mainly includes Hong Kong, Shenzhen, Dongguan, and Zhuhai, which have coordinated development degrees ranging between 0.0 and 0.135 and the highest emergy density (ED) values but are extremely dependent on external emergy. They have high levels of emergy use per capita (EUC), high living standards, and high quality of life. The effective use of land resources severely restricts sustainable economic development, resulting in extreme ecological and environmental carrying pressure. Category 2 includes Guangzhou, Macau, Foshan, and Zhongshan, whose coordinated development degrees range from 0.143 to 0.179. The sustainable development capacity of these cities is at the middle level amongst the whole GBA. Their main emergy characteristics are emergy flow and subsystem evaluation indices that are between category 1 and category 3, but each has its own characteristics. The category 3 cities include Zhaoqing, Jiangmen, and Huizhou, whose coordinated development degrees are between 0.192 and 0.369. These cities are characterized by relatively low ED and EUC values, living standards, and quality, but their land resources have certain potential. These cities have a high emergy self-sufficiency rate (ESR) and natural environmental support capacity, but their environmental loading ratio (ELR) is still much higher than the national average. In terms of the economic development and innovation development levels, these cities are ranked as category 1 > category 2 > category 3. In terms of the ecological and environmental conditions and





blue–green space protection, these cities are ranked as category 1 < category 2 < category 3. The results of this study can provide cities in the GBA with more scientific and consistent directions for the coordinated development of their ecological–economic–social systems to provide sustainable development decision-making services for megacity cluster systems.

**Keywords:** coordinated development degree; entropy method; emergy; megacity cluster; GBA

## 1. Introduction

A city cluster is a collection of cities and towns of different sizes and functions with independent and closely related ecosystems and economic systems, concentrated in a certain area with a high density of emergy flows [1], which originated from the concept of the megalopolis proposed by the French geographer J. Gottmann in 1957 [2]. City clusters have been widely studied by domestic and foreign academics and have been formulated in different ways. American scholars call the concept the global city region [3], British scholars call it the megacity region [4,5], and Chinese scholars such as Yao Shimou and Fang Chuanglin were the first to use the concept of city clusters in China [6,7].

Huang et al. studied the spatial distribution of 273 cities in China and measured city eco-efficiency levels in China from 2003 to 2015. Their empirical study showed that city clusters and eco-efficiency have generally increased in China, and the increase in city clusters is conducive to improving city eco-efficiency [8]. However, natural and human-made factors lead to natural and artificial ecological risks that break the ecological balance relationship of city clusters. With the impacts of extreme climate disasters, such as global warming, sea level rise, and typhoons, coupled with the continuous acceleration of urbanization, the construction of high-density city clusters, and pollution and wastewater being discharged, water quality and environmental quality are severely degraded; ecological self-regulation is severely weakened; and ecosystems such as rivers, lakes, and seashores are heavily damaged [9–11]. At present, the issue of how to coordinate the relationship between urbanization development and ecological environmental protection is a common concern among academics and government policy makers, and it is a major difficulty that needs to be solved and has risen to a worldwide strategic issue. As an open social–economic–natural complex megasystem [12], it is urgent to study the sustainable development of the ecological and economic systems and their optimal countermeasures, but the corresponding research is still weak [7].

The main measurement tools for evaluating the degree of sustainability and city health are city sustainability evaluation indicators and methods, among which, the emergy analysis method is of particular interest. In the 1980s, H.T. Odum, a famous American ecologist, created the emergy theory and analysis method [13–15], which was introduced and developed greatly in China in the 1990s [16,17]. An emergy analysis is based on eco-thermodynamics and used to measure all products and service activities based on solar emergy. The unit is the solar joule (sej), and any material and emergy have a corresponding conversion rate (sej/j). The conversion rate converts the emergy flow, monetary flow, population flow, and information flow of the system into a unified standard solar value for measurement and analysis, and it comprehensively evaluates the system structure, functional characteristics, and ecological and economic benefits [18]. The emergy analysis method solves all the different categories of emergy, resources, products, and even labor and services of various types that are incomparable and difficult to account for, which is a major breakthrough in both theory and method [19].

In recent years, the structure, function, and sustainable development of eco-economic systems at different scales have been studied at home and abroad using emergy analysis methods for sustainable development [1,5,17,19–22], and the studied cities include Qingdao, Shangluo, Puerto Rico, Ningxia, Haixi, Wuhan, and other cities or regions [8,23–26]. A large number of studies have quantitatively studied the ecological and economic environment

and sustainable development of the Shandong coastal city cluster, Chengdu–Chongqing city cluster, Changzhutan city cluster, Beijing–Tianjin–Hebei city cluster, Taihu Lake city cluster, and Yangtze River city cluster in the Yangtze River Delta [6,20,27–30]. Some scholars have explored in depth the theoretical [31], methodological [32,33], and benchmarking [13–15] aspects, especially the calculation methods of emergy, such as the complexity of city import and export products, emergy conversion rate data, confusion in the use of emergy baselines, and the lack of separation of labor and services from products in the emergy conversion rate, making the results difficult to compare. Deduplicating calculations [16], improvements to the emergy ecological footprint model [34] and other issues have been discussed. However, the study of sustainable development of high-density megacity clusters based on the emergy analysis method still needs further improvement.

The current research based on emergy analyses combines several eco-economic system indicators, such as the emergy sustainable index (ESI) and emergy index for sustainable development (EISD), as measures of the sustainability and health of eco-economic systems in city clusters. The empirical evidence shows that this approach is not applicable for economically developed cities [16,21], as these indices are inevitably not comprehensive and accurate due to the combination of only some indicators of city eco-economic systems. The GBA city cluster is one of the five megacity clusters in China, a strategic core area of national economic development and the main area of national new urbanization. It is facing the same threats of unsustainable resources and environmental security as other megacity clusters, such as high-density clusters, high-rate expansion, high-intensity pollution, and high risk [9]. In view of this, this study takes the GBA city cluster as an example, and on the basis of the emergy analysis theory, we propose a new method to evaluate the sustainable development capacity using all indicators of the ecological and economic system of the city cluster and propose an evaluation of the sustainable development capacity of the city cluster using a development coordination analysis, which makes the evaluation more accurate, systematic, and comprehensive. This study can provide a scientific reference and decision-making basis for the systems planning of the sustainable development of the city cluster.

## 2. Materials and Methods

### 2.1. Research Ideas

The research ideas and process used in this study are shown in Figure 1.

### 2.2. Overview of the GBA

The GBA is located in South China ($21°25'$ N–$24°30'$ N, $111°12'$ E–$115°35'$ E), and as a specific geographical unit, it includes a coastal area consisting of several connected bays and harbors with coastlines recessed inland, as well as land areas bordering the bays or harbors and adjacent islands, bridging Hong Kong, the Macau Special Administrative Region and Guangdong Province in Guangzhou, Shenzhen, Zhuhai, Foshan, Huizhou, Dongguan, Zhongshan, Jiangmen, Zhaoqing, and many other town clusters distributed in ports or inlets of the sea [18,35,36]. With a total area of 56,000 km$^2$, the population of the GBA reached 86 million by the end of 2020, making it a typical high-density town cluster and high-density population area, being equivalent to 3 times that of the New York Bay Area, 1.5 times that of the Tokyo Bay Area, and nearly 10 times that of the San Francisco Bay Area, with the characteristics of high population density, a high proportion of city dwellers, and a large proportion of foreign residents. It is an important strategic space where the two trade circles of the Pacific Rim and the Indian Ocean Rim are combined, assuming the important task of leading the transfer of trade from the Pacific Rim to the Indian Ocean Rim [37], having an important strategic position in overall national development. Due to the differences in the gradients of the elements and the combination of elements at each level, the GBA has actually become an interlocking composite ecosystem composed of various landscapes, including water and land areas (high-density city area, terrestrial forest areas, river area, palletized fruit forest areas, other agricultural production area, estuarine

runoff-dominated area, mixed freshwater and seawater area, outer seawater-dominated areas, and other ecosystems covering water and land). The ecosystem covers a wide range of ecosystem services, including support services such as biological habitat, biodiversity, and primary habitat services for marginal species; provisioning services such as food and water supply, gene pool, and water balance services; regulating services such as extreme climate regulation, water purification, and flood control and storage services; and various ecosystem services such as ecological aesthetics, cultural education, and recreation services. With rapid economic development, the GBA is now facing the dual impacts of natural ecological risks and anthropogenic ecological risks, and the loss of ecological space is obvious. The expansion of city construction land has led to the decline of the structure and function of the regional natural ecosystem, which has seriously affected the sustainable development of the regional ecological and economic system.

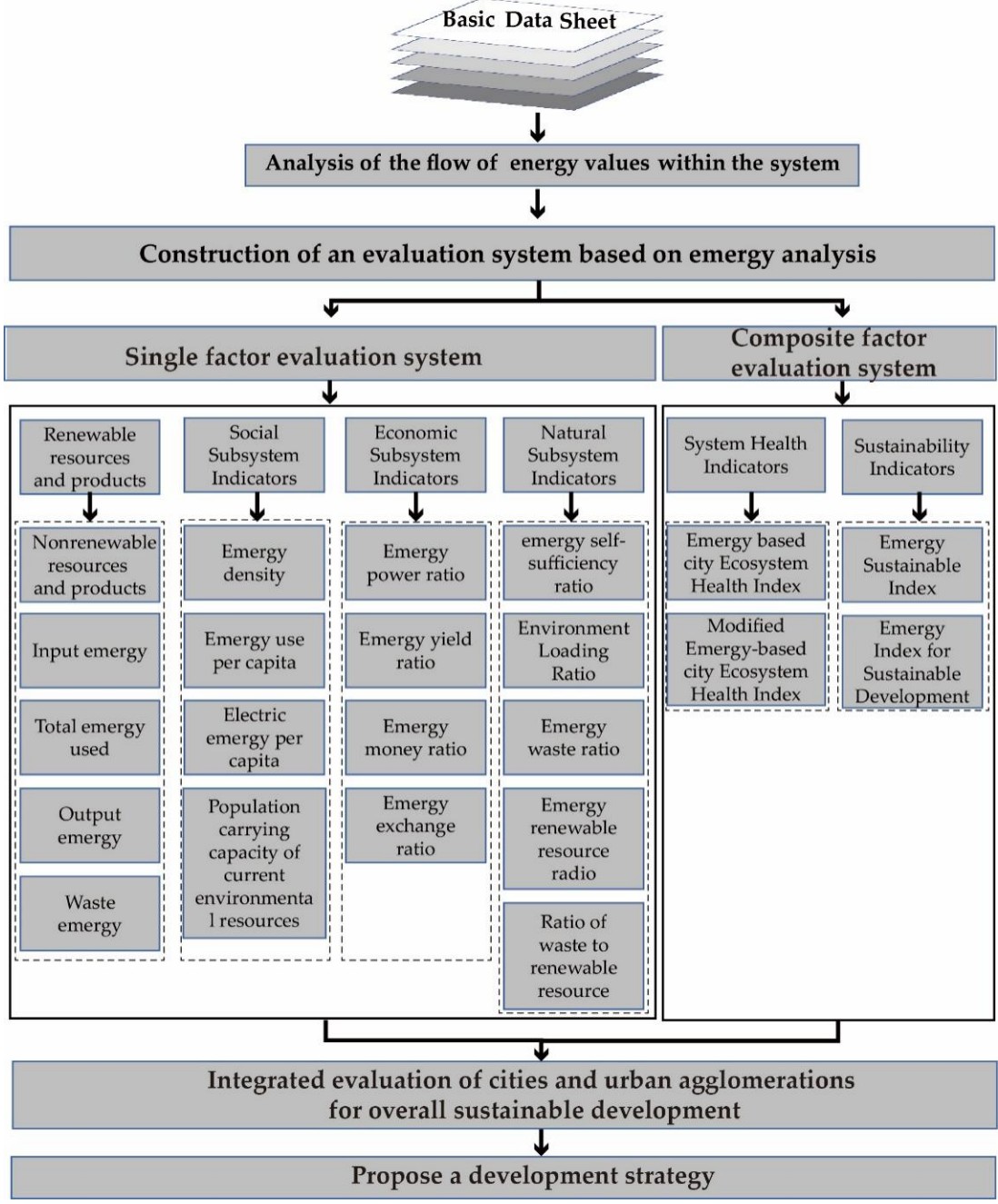

**Figure 1.** Article flowchart.

### 2.3. Data Sources

The data used in this study to analyze the emergy inputs and outputs of the eco-economic system of the GBA city cluster mainly included the statistical reports for China and the yearbooks of the cities in the GBA in 2019, the statistical bulletin on national economic and social development, and other information on the natural environment, geography, economy, and society.

### 2.4. Accounting for the Emergy of the GBA

The research idea is that first we will draw an emergy system map and conduct an emergy analysis on the basis of data collection to establish an emergy indicator evaluation system for city clusters in the GBA; second, we will analyze the sustainable development of the city cluster's eco-economic system; finally, we will propose a new method to evaluate the sustainable development capacity using all indicators of the city cluster's eco-economic system and evaluate the sustainable development capacity of city clusters in combination with the degree of coordinated development.

The ecological and economic system of the GBA is mainly composed of forest ecosystems; freshwater ecosystems such as wetlands, rivers, and basin ponds; marine ecosystems; agricultural ecosystems; rural ecological and economic systems; and ecological and economic subsystems such as cities, towns, and villages, while eleven cities consisting of Guangzhou, Shenzhen, Hong Kong and Macau and their surrounding towns and villages together constitute a collection of interrelated and coupled city clusters (Figure 1). In the system, in the form of emergy flow, renewable emergy sources such as solar emergy, wind emergy, potential emergy, and chemical emergy from rainwater, as well as geocyclonic emergy, are absorbed by forests and water ecosystems; nonrenewable emergy sources such as raw coal, fuel oil, and tidal emergy form water and electricity to supply the city clusters, for rural production and living, and for manufacturing and other industries. In the form of material flow, the population flow and monetary flow from decentralized rural sources are invested into the system. Based on the form of material flow, agriculture, fishery, animal husbandry, and other industries produce grain, aquatic products, meat, fruits, and vegetables to supply to city clusters, towns, and cities, and even for export to provincial, foreign, and international markets. Nonrenewable resources such as minerals are transformed into metals to enter the mining industry and finally flow into the manufacturing industry. Based on the form of capital flow and population flow, the capital flow and population flow of city clusters are injected into manufacturing, financial, commercial, and service industries, as well as other kinds of industries, and the output material flow, money flow, and population flow form a close relationship with the province and the outside areas, at home and abroad. The international market injects material flow, money flow, and population flow into the ecological and economic system of the GBA through imported goods, imported services, and inbound tourism. There are also close population, material, and money flows between the cities within the city cluster and the surrounding scattered rural sources, and the generated waste is released back into the natural environment, such as to forests, waters, and oceans, through the waste stream. The emergy flows through the eco-economic system in many forms, which forms a complex eco-economic megasystem of interdependence and mutual feedback (Figure 2).

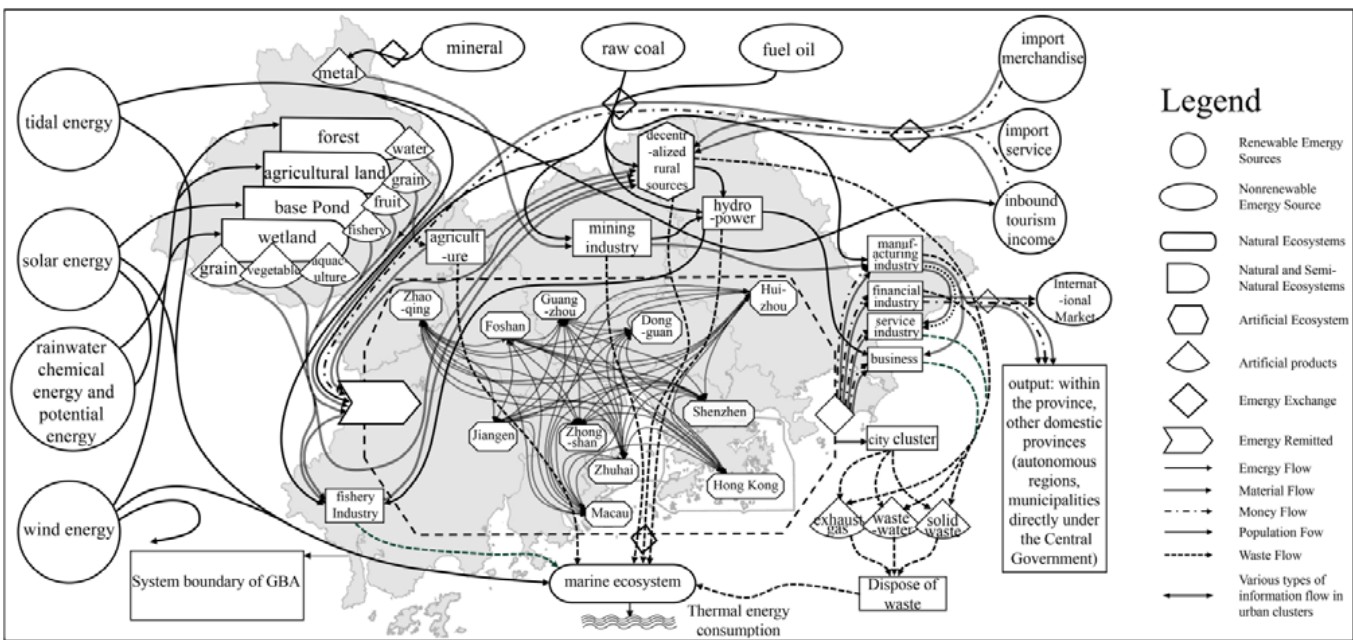

**Figure 2.** Schematic diagram of the GBA emergy system.

Based on Figure 1, we established an emergy index system with reference to several literature choices [20,38,39], including six subindex systems of the system emergy flow index; a sustainability index; a system health index; and social, economic, and natural subsystem evaluation indices. As shown in Table 1, there are six system emergy flow indicators, such as renewable resources and product emergy and nonrenewable resources and product emergy. The social subsystem evaluation indicators include 4 indicators, which mainly reflect the population-carrying capacity of a city and discriminate the use types of emergy [1], such as ED and EUC. The economic subsystem evaluation indicators include 4 indicators, which reflect the regional economic development and the rationality of its structure [30], such as the emergy power ratio (EPR) and emergy money ratio (EMR). The natural subsystem evaluation indicators include five indicators of the intensity of the anthropogenic disturbance to the ecological environment and the emergy yield capacity of each city, which indicate the intensity of anthropogenic disturbance to the ecological environment and emergy yield capacity [23], such as the emergy self-sufficiency ratio (ESR) and environment loading ratio (ELR). In particular, it is noted that the input emergy rate is inversely related to the output rate and is not an independent indicator; therefore, the emergy–value input rate is excluded from the indicator system [40]. We use the emergy benchmark value of $15.83 \times 10^{24}$ seJ/a [13], as modified by Odum in 2000, to calculate the main emergy, material, and monetary flows of the system. The emergy or money flows in different units of measure (J, g, or USD) are converted into emergy units (sej) according to the corresponding emergy conversion rates of each resource.

**Table 1.** The eco-economic emergy index system and analysis of the GBA.

| Indicator System | Index | Symbol (Unit) | Expression for Computation | Hong Kong | Macau | Guangzhou | Shenzhen | Zhuhai | Foshan | Huizhou | Dongguan | Zhongshan | Jiangmen | Zhaoqing |
|---|---|---|---|---|---|---|---|---|---|---|---|---|---|---|
| **Emergy Flow Indicators** | Renewable resources and products | $\times 10^{18}$ R (sej) | R | 5.20 | 0.15 | 32.80 | 9.44 | 7.89 | 16.50 | 71.00 | 12.00 | 8.28 | 42.10 | 64.70 |
| | Nonrenewable resources and products | $\times 10^{22}$ N (sej) | N | 8.69 | 0.34 | 14.90 | 6.62 | 2.01 | 3.40 | 3.55 | 5.97 | 2.06 | 3.66 | 2.04 |
| | Input emergy | $\times 10^{23}$ I (sej) | I | 605.00 | 2.29 | 3.11 | 8.95 | 0.10 | 0.75 | 0.99 | 3.49 | 0.369 | 0.039 | 0.11 |
| | Total emergy used | $\times 10^{23}$ U (sej) | U = R + N + I | 606.00 | 2.33 | 4.74 | 9.73 | 1.21 | 1.16 | 1.39 | 4.15 | 0.06 | 0.71 | 0.36 |
| | Output emergy | $\times 10^{23}$ O (sej) | O | 886.00 | 0.06 | 1.16 | 6.54 | 0.80 | 0.42 | 0.75 | 4.00 | 0.30 | 0.28 | 0.01 |
| | Waste emergy | $\times 10^{21}$ W (sej) | W | 7.42 | 0.67 | 14.80 | 15.80 | 5.38 | 8.95 | 0.400 | 12.70 | 3.58 | 5.18 | 0.36 |
| **Social Subsystem Indicators** | Emergy density | $\times 10^{15}$ ED (sej/m$^2$) | U/A | 54.80 | 7.08 | 638.00 | 48.70 | 697.00 | 305.00 | 0.01 | 0.16 | 0.03 | 0.001 | 0.002 |
| | Emergy use per capita | $\times 10^{16}$ EUC (sej/Person) | U/P | 814.00 | 34.90 | 3.18 | 7.47 | 6.40 | 1.47 | 2.88 | 4.95 | 1.82 | 1.54 | 1.07 |
| | Population-carrying capacity of current environmental resources | $\times 10^{2}$ PCC (ten thousand people) | (R + I)/U·P | 7.44 | 0.66 | 9.78 | 12.00 | 1.56 | 5.13 | 3.45 | 7.04 | 2.01 | 2.38 | 1.07 |
| | Electric emergy per capita | $\times 10^{15}$ EEC (sej/Person) | Electric/P | 3.42 | 4.81 | 3.62 | 4.01 | 5.36 | 2.13 | 4.87 | 5.54 | 5.05 | 3.53 | 2.98 |
| **Economic Subsystem Indicators** | Emergy power ratio | $\times 10^{-1}$ EPR | Electric/U | 0.004 | 0.14 | 1.14 | 0.54 | 0.84 | 1.45 | 1.69 | 1.12 | 2.81 | 2.28 | 2.77 |
| | Emergy yield ratio | EYR | EYR = U/I | 1.00 | 1.02 | 1.52 | 1.09 | 1.21 | 1.54 | 1.40 | 1.19 | 1.65 | 2.36 | 3.13 |
| | Emergy money ratio | $\times 10^{12}$ EMR (sej/USD) | U/GDP | 21.30 | 4.23 | 1.37 | 2.66 | 2.76 | 0.78 | 2.24 | 3.32 | 1.09 | 1.62 | 25.10 |
| | Emergy exchange ratio | EER | I/O | 0.68 | 35.50 | 2.67 | 1.37 | 1.26 | 1.81 | 1.32 | 0.87 | 1.22 | 1.09 | 12.60 |
| **Natural Subsystem Indicators** | Emergy self-sufficiency ratio | ESR | (R + N)/U | 0.001 | 0.01 | 0.31 | 0.07 | 0.17 | 0.29 | 0.26 | 0.14 | 0.34 | 0.52 | 0.57 |
| | Environment loading ratio | $\times 10^{4}$ ELR | (I + N)/R | 1170.00 | 155.00 | 1.45 | 10.30 | 1.53 | 0.70 | 0.20 | 3.46 | 0.73 | 0.17 | 0.06 |
| | Emergy waste ratio | $\times 10^{-2}$ EWR | W/U | 0.01 | 0.29 | 3.12 | 1.62 | 4.45 | 7.72 | 2.91 | 3.06 | 5.96 | 7.31 | 10.10 |
| | Emergy renewable resource radio | $\times 10^{-5}$ ERR | R/U | 0.001 | 0.06 | 6.92 | 0.97 | 6.52 | 14.20 | 51.10 | 0.03 | 13.80 | 59.40 | 181.00 |
| | Ratio of waste to renewable resource | $\times 10^{2}$ EWI | W/R | 14.30 | 44.90 | 4.51 | 16.70 | 6.820 | 5.420 | 0.57 | 10.60 | 4.32 | 1.23 | 0.56 |

**Table 1.** *Cont.*

| Indicator System | Index | Symbol (Unit) | Expression for Computation | Hong Kong | Macau | Guangzhou | Shenzhen | Zhuhai | Foshan | Huizhou | Dongguan | Zhongshan | Jiangmen | Zhaoqing |
|---|---|---|---|---|---|---|---|---|---|---|---|---|---|---|
| **Sustainability Indicators** | Emergy sustainable index | $\times 10^{-6}$ ESI | EYR/ELR | 0.09 | 0.07 | 105.00 | 10.50 | 79.20 | 219.00 | 716.00 | 34.50 | 228.00 | 1400.00 | 569.00 |
| | Emergy index for sustainable development | $\times 10^{-5}$ EISD | (EYRX EER)/ (ELR + EWI) | 0.006 | 2.32 | 37.30 | 1.42 | 9.54 | 36.80 | 92.10 | 2.91 | 26.20 | 142.00 | 6520.00 |
| **System Health Indicators** | Emergy-based city ecosystem health index | $\times 10^{3}$ EUEHI | EYRX EERXED)/(ELRX EMR) | 0.15 | 38.80 | 12.70 | 2.60 | 2.41 | 14.50 | 3.60 | 1.43 | 8.11 | 6.55 | 6180.00 |
| | Modified emergy-based city ecosystem health index | $\times 10^{-10}$ EUEHI′ | (EYRX EERX ESR)/ (ELRX EWI) | 0.0006 | 0.75 | 1970.00 | 5.87 | 243.00 | 2140.00 | 4,260,000.00 | 40.80 | 221.00 | 64,100.00 | 739,000,000.00 |

P: Population size; A: land area; E: electricity consumption emergy; Person: the people. Note: The meaning of each indicator above can be found in the literature [9,20].

*2.5. Entropy Method*

The existing sustainability evaluations are all combinations of a few evaluation indicators, such as the ESI and EISD. Although the data are easy to obtain and the results have some ecological and economic significance, the number of selected indicators is so small that the results may even appear to be completely opposite and uninterpretable. For the first time, we propose using the full range of indicators of the complete socioeconomic–natural subsystem to evaluate the sustainable development capacity. The entropy method can avoid the disadvantages of the subjective selection of index weights. Entropy was first introduced into information theory by Shenon and has been widely used in engineering, technology, social economy, and other fields [41]. The basic idea of the entropy weighting method is to determine objective weights based on the magnitude of the variability of the indicators. Generally, if the information entropy, $E_j$, of an indicator is smaller, it indicates that the degree of variation of the indicator value is larger, more information is provided, the role it can play in the comprehensive evaluation is larger, and its weight is also larger. In contrast, the greater the information entropy of an indicator, the less variation in the indicator value, the less information provided, the smaller the role played in the comprehensive evaluation, and the smaller the weight. The entropy method assigns weights in the following steps.

2.5.1. Data Standardization

First, the initial data are standardized, assuming that the number of cities to be evaluated is m, the evaluation indicators equal n, and the original data of the jth indicator of the ith city equal xij (i = 1, 2 ... m; j = 1, 2 ... n).

When the data are positively correlated with sustainability,

$$y_{ij} = \frac{x_{ij} - \min(x_{1j}, \ldots x_{mj})}{\max(x_{1j}, \ldots x_{mj}) - \min(x_{1j}, \ldots x_{mj})} \tag{1}$$

When the data are negatively correlated with sustainability,

$$y_{ij} = \frac{\max(x_{1j}, \ldots x_{mj}) - x_{ij}}{\max(x_{1j}, \ldots x_{mj}) - \min(x_{1j}, \ldots x_{mj})} \tag{2}$$

where $y_{ij}$ represents the standardized data, $x_{ij}$ represents the original data, $\max(x_{1j}, \ldots x_{mj})$ is the maximum value of each index dataset, and $\min(x_{1j}, \ldots x_{mj})$ is the minimum value of each index dataset.

2.5.2. Indicator Information Entropy Value $E_j$

$$E_j = -\frac{1}{\ln m}\sum_{i=1}^{m} p_{ij} \ln p_{ij} (i = 1, 2, \ldots m; \ j = 1, 2, \ldots n) \tag{3}$$

Here, A is the weight of the standardized indicator of the ith city under the j indicator. If $p_{ij} = 0$, define $E_j = 0$.

The entropy weight is calculated from the information entropy calculation of each evaluation index, and it is used to evaluate the degree of difference between the values of each index in the index system. A larger entropy weight, $w_j$, indicates a more important position of the indicator in the evaluation. Formula (4) is as follows:

$$w_j = \frac{1 - E_j}{n - \sum_1^n E_j} \tag{4}$$

Here, $E_j$ is the information entropy value of the jth indicator of the ith city.

2.5.3. System Evaluation

The evaluation value of the ith city:

$$f_i = \sum_{j=1}^{n} w_j y_{ij} \ (i = 1, 2, \ldots m) \tag{5}$$

Here, the higher the evaluation value $f_i$, the more sustainable it is.

*2.6. Evaluation of the Degree of System Coordination Development*

The ecological–economic system of a city cluster is a complex ecosystem consisting of socioeconomic–natural subsystems. In the process of the dynamic interaction of the three subsystems of the city system, two situations may occur: in one case, the three subsystems continuously coordinate to jointly promote city development; in the other case, one or several subsystems are not coordinated with other subsystems, which affects the development of other subsystems or the whole city system or even leads to the destruction of the normal structure and function of the city system. Thus, it is indispensable to study the development of the system while considering whether the system is coordinated or not.

At present, the fuzzy comprehensive evaluation method, gray system model method, and coupled coordination degree model are commonly used to construct sustainable development index system methods. In this study, the coupled coordination degree model is used. According to the systems theory, the higher the degree of coordination between systems, the smaller the coefficient of variation. The equation for the degree of coordination is as follows:

$$C = \prod_{k=1}^{3} E_k / \left[ \sum_{k=1}^{3} \left( \frac{E_k}{3} \right) \right]^3 \tag{6}$$

where $E_k$ (k = 1,2,3) is the sustainable development index of the social, economic, and natural subsystems, and Equation (6) is essentially the ratio of the geometric mean to the arithmetic mean of the sustainable development index of the three subsystems. The greater the degree of coordination C, the better the coupling between the subsystems, but the degree of coordination C still cannot express the coordination of the development of the whole system. We define the developmental coherence as follows:

$$D = (C \times T)^{1/2} \tag{7}$$

where T in the formula is the weighted average of the 3 subsystem sustainability indices:

$$T = \sum_{k=1}^{3} \alpha_k E_K \tag{8}$$

In this paper, k = 1, 2, and 3 represent the social, economic, and natural subsystems, respectively, because the entropy value method is used to calculate the weights for the whole ecological and economic system without additional consideration of the weights; therefore, the weights are taken here. The development coordination degree covers the coupling degree and the sustainable development index of the whole system, which can comprehensively reflect the development and coupling situation of the whole system.

**3. Results and Analysis**

Based on the calculation results in Table 1, we can analyze the emergy flow and the three subsystems of the social–economic–natural system and the whole eco-economic system of the GBA city cluster.

*3.1. System Emergy Flow Analysis*

The total consumption emergy U values of most cities in the GBA are in the $10^{22}$ sej to $10^{23}$ sej level, among which, Macau, Guangzhou, Shenzhen, Zhuhai, Foshan, Huizhou, and Dongguan are in the $10^{23}$ sej level and Zhongshan, Jiangmen, and Zhaoqing are in the $10^{22}$ sej level, except for Hong Kong, in which the U value is as high as $6.06 \times 10^{25}$ sej,

and the U value is higher than the other value. The U value is two ranks higher than the other 10 cities and cannot be compared on one graph; therefore, the data for the 10 cities except for Hong Kong are marked out in the left and middle graphs in Figure 3, but Hong Kong is still included in the numerical analysis. The reason for this is that the renewable resources and product emergy R of the GBA city cluster are only $10^{17}$ sej~$10^{19}$ sej orders of magnitude different; the lowest in Macau is only $10^{17}$ sej, and Hong Kong and Macau are at $10^{18}$ sej, which are both lower in proportion. The emergy levels of nonrenewable resources and products N in order of quantity are Guangzhou, Hong Kong, Shenzhen, Dongguan, Jiangmen, Huizhou, Foshan, Zhongshan, Zhaoqing, Zhuhai, and Macau, most of which are $10^{22}$ sej; only Macau is the lowest at $10^{21}$ sej, while the import emergy I is $10^{22}$ sej~$10^{23}$ sej in most cities except Hong Kong. The order of magnitude in Macau, Guangzhou, and Shenzhen reaches $10^{23}$ sej, while Hong Kong's high import emergy reaches $10^{25}$ sej, indicating that the large amount of import emergy is crucial to the impact of Hong Kong's ecological and economic system. The results show that Macau has the smallest area and the smallest population, but its U value exceeds those of Huizhou, Zhuhai, Foshan, Jiangmen, Zhongshan, and Zhaoqing. Overall, the emergy of renewable resources and products in the GBA city cluster is only $10^{19}$ sej orders of magnitude, which accounts for a very low proportion of the total consumed emergy, U. The difference between R and U is 3–7 orders of magnitude, among which Hong Kong, Macau, Shenzhen, and Dongguan have the largest differences, indicating that the structure of the total consumed emergy in the GBA city cluster is extremely unreasonable, relying more on the imported emergy and emergy of nonrenewable resources and products. Combining imports and exports into monetary flows for the analysis, Hong Kong's import emergy is as high as $6.06 \times 10^{25}$ sej, which is 2–3 orders of magnitude higher than that of other cities, with 31.6% of imported goods, 67.9% of actual foreign capital utilization, and 0.5% of inbound tourism income. On the other hand, Hong Kong's export emergy is as high as $8.86 \times 10^{25}$ sej, indicating that Hong Kong's eco-economic system is highly dependent on import emergy and export emergy. The import emergy levels of Macau and Zhaoqing are both two orders of magnitude higher than the export emergy and have not reached equilibrium; the rest of the cities are of the same order of magnitude and are more balanced. The waste emergy levels of the 11 cities in the GBA are of the order of $10^{21}$ sej to $10^{22}$ sej, all greatly exceeding the renewable resource and product emergy levels; among them, Shenzhen, Guangzhou, and Dongguan exceed the extraordinary values, followed by Foshan, Hong Kong, and Jiangmen, indicating that the cities in the GBA have low utilization rates of renewable resources and recycling rates of waste (Table 1, Figure 3).

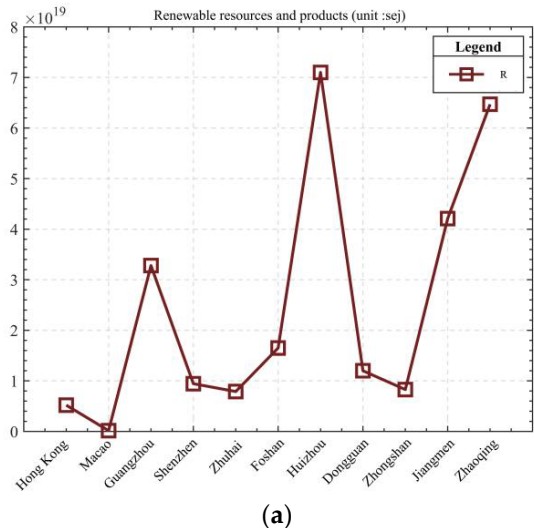

(a)

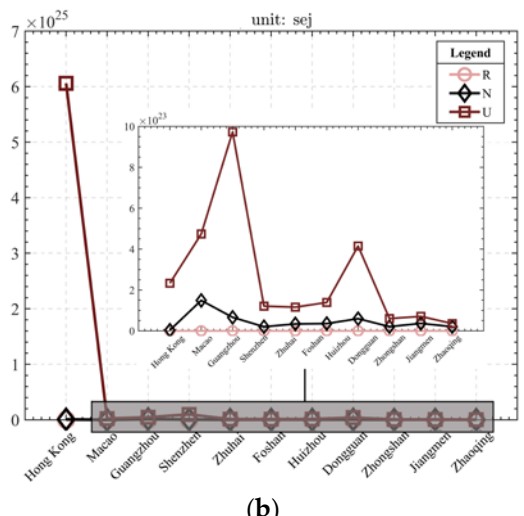

(b)

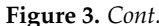

**Figure 3.** *Cont.*

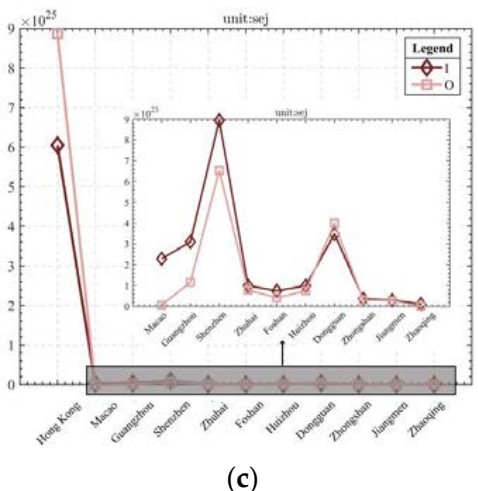

(**c**)

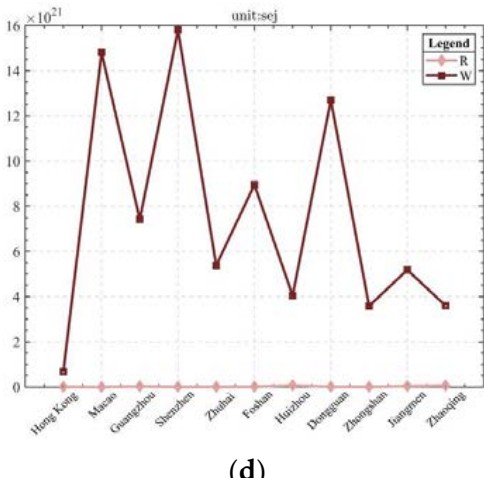

(**d**)

**Figure 3.** Emergy flow index of the GBA (**a**) The GBA's comparative map of renewable natural resources and products by citys; (**b**) Comparison of renewable natural resources and products (R), Nonrenewable natural resources and products (N) and Total emergy use (U) by citys in The GBA; (**c**) Comparison of input emergy (I) and output emergy (O) for each citys in The GBA; (**d**) Comparison of renewable natural resources and products (R) and emergy of waste (W) by citys in The GBA.

### 3.2. Social Subsystem Emergy Analysis

The national average ED at $20^{16}$ is $6.58 \times 10^{12}$ sej/m$^2$ [12]. The ED of Hong Kong is as high as $5.48 \times 10^{16}$ sej/m$^2$ and that of Macau is as high as $7.08 \times 10^{15}$ (sej/m$^2$). This is one to four orders of magnitude higher than those of the remaining nine cities and far exceeds the national average. This indicates that the sustainable development of eco-economic systems in Hong Kong and Macau is under great pressure from land scarcity. The remaining cities in order of magnitude are Shenzhen, Dongguan, Zhuhai, Guangzhou, Zhongshan, and Foshan. The values for these cities are all one order of magnitude higher than the national average, indicating that the sustainable development of the eco-economic system in these cities is also under pressure from land scarcity, while Huizhou, Jiangmen, and Zhaoqing are of the same order of magnitude as the national average, and Zhaoqing is also lower than the national average, indicating that Zhaoqing has the potential for sustainable land development. In contrast, Hong Kong's EEC is as high as $8.14 \times 10^{18}$ sej/person and Macau's EEC is as high as $3.49 \times 10^{17}$ sej/person, which is one to two orders of magnitude higher than those of the remaining nine cities and far exceeds the national average of $4.57 \times 10^{16}$ sej/person [42]. Next, Shenzhen, Zhuhai, and Dongguan are higher than the national average, while Guangzhou, Huizhou, Zhongshan, Jiangmen, Foshan, and Zhaoqing are lower than the national average. The EEC levels in order of quantity are Dongguan, Zhuhai, Zhongshan, Huizhou, Macau, Shenzhen, Guangzhou, Jiangmen, Hong Kong, Zhaoqing, and Foshan. The relatively high EEC of the GBA city cluster indicates a high level of industrial technology informatization. In terms of the population that can be carried under the current environmental level, except for Macau, where the population-carrying capacity and the actual population are essentially balanced due to the high value of imported emergy, the actual population of the remaining 10 cities is much higher than the population-carrying capacity, especially for Shenzhen, Guangzhou, and Hong Kong, followed by Dongguan and Foshan, indicating a higher population-carrying pressure (Table 1, Figure 4).

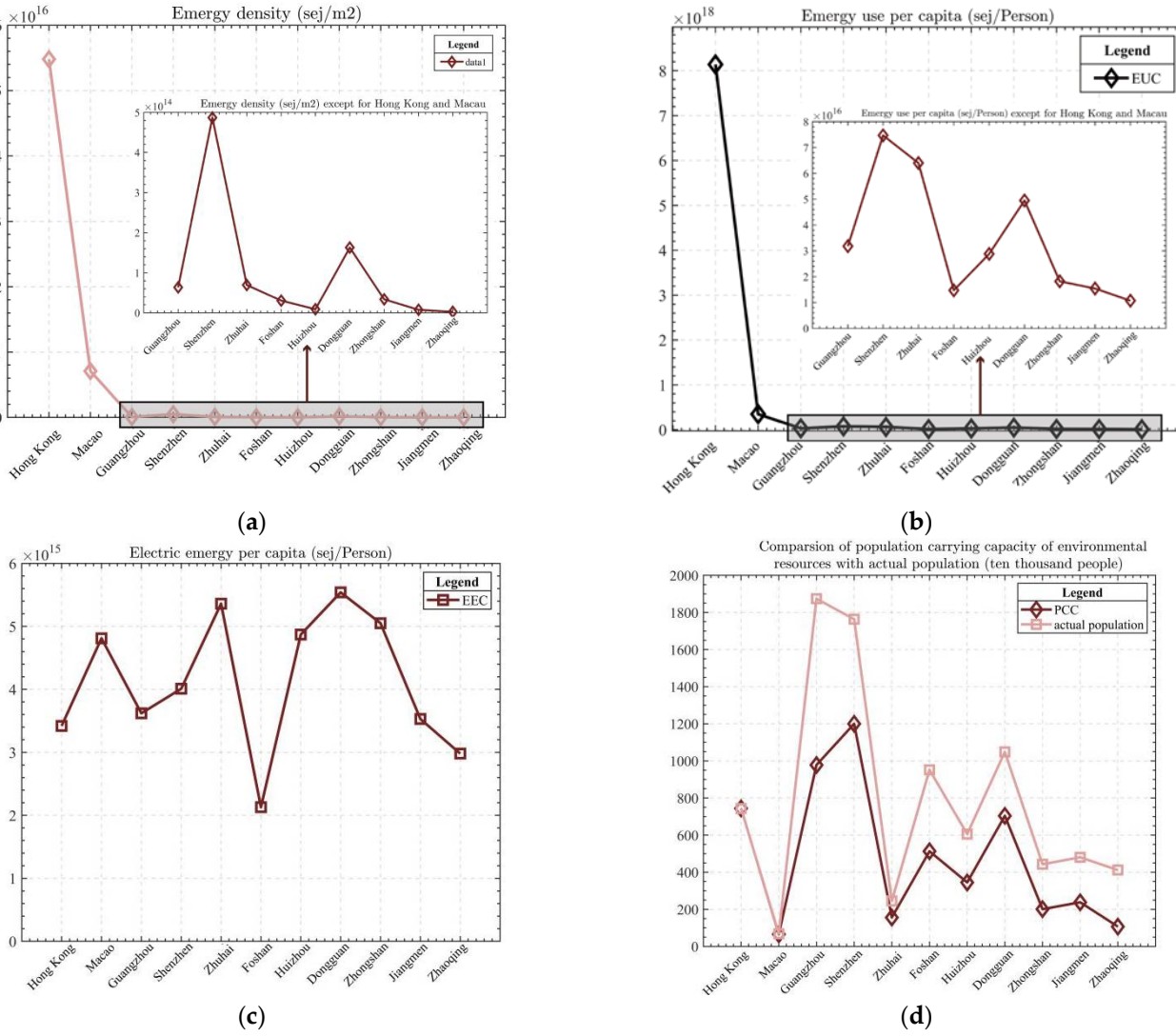

**Figure 4.** Social subsystem emergy index (**a**) Comparison of emergy density (ED) by city in the GBA; (**b**) Comparison of Emergy use per capita (EUC) by citys in the GBA; (**c**) Comparison of Electric emergy per capita (EEC) by citys in the GBA; (**d**) Comparison of Population carrying capacity of current environmental resources (ten thousand people) (PCC) and actual population by citys in the GBA.

### 3.3. Economic Subsystem Emergy Analysis

The electric emergy per capita (EPR) levels by order of magnitude are Zhongshan, Zhaoqing, Jiangmen, Huizhou, Foshan, Guangzhou, Dongguan, Zhuhai, Shenzhen, Macau, and Hong Kong. The total emergy use in Hong Kong and Macau is much higher by an order of magnitude relative to the electric emergy, meaning the electric emergy ratio is almost zero.

The emergy yield ratio (EYR) levels in order of quantity are Zhaoqing, Jiangmen, Zhongshan, Foshan, Huizhou, Zhuhai, Dongguan, Shenzhen, Macau, and Hong Kong; the emergy output rate of Hong Kong is 1, while Macau and Shenzhen are only slightly greater than 1. This situation is due to the high proportion of imported emergy to the total emergy used in the three cities, which once again shows the dependence of these cities on imported emergy and the important role of imported emergy in the emergy analysis of this case. Conversely, Zhaoqing has a low ratio of imported emergy to total emergy used and a relatively high emergy of renewable resources and products, making the emergy output ratio higher. The EMR is the ratio of the emergy input that supports the operation

of a country or region's eco-economic system to the country or region's GDP, which can be understood as the amount of emergy that can be purchased per unit of money. Hong Kong has the highest EMR of $2.13 \times 10^{13}$ sej/USD, while Zhaoqing has the lowest EMR of $2.51 \times 10^{11}$ sej/USD, with a difference of two orders of magnitude between them. Hong Kong is well above the 2016 Chinese average emergy money rate of $5.65 \times 10^{12}$ sej/USD [12], while all other cities in the GBA are below the national average emergy money rate. This indicates that the same money can buy more emergy in Hong Kong, and despite the high EMR in Hong Kong, the vast majority of the export emergy is generated by re-export trade, while Hong Kong does not lose more resources. The largest emergy exchange rate in Macau indicates that the imported emergy is much larger than the exported emergy, which is due to the special economic structure of Macau and is in line with the way in which developed regions obtain much more emergy from outside than the exported emergy, followed by Zhaoqing, indicating that the imported emergy of Macau and Zhaoqing is larger than the exported emergy. The EER of Hong Kong and Dongguan is <1, indicating that the export emergy of Hong Kong and Dongguan is larger than the import emergy, and the re-export trade of the cities does not lose more resources. The rest of the cities have a value of 1 < EER< 10, indicating that their import emergy is greater than their export emergy, which is also more in line with the economic characteristics of developed regions (Table 1, Figure 5)

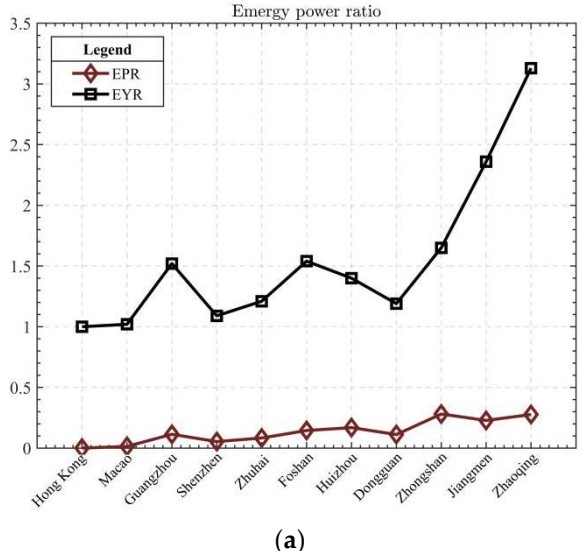

(a)

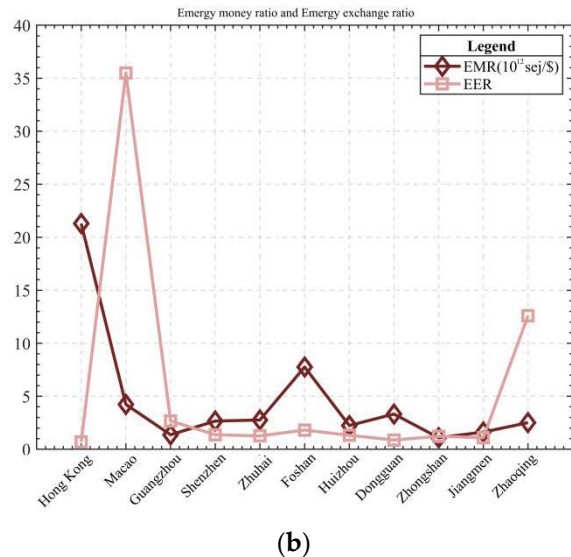

(b)

**Figure 5.** Economic subsystem emergy index (**a**) Comparison of Emergy power ratio (EPR) and Emergy yield ratio (EYR) by cities in the GBA; (**b**) Comparison of Emergy money ratio (EMR) and Emergy exchange ratio (EER) by cities in the GBA.

*3.4. Natural Subsystem Emergy Analysis*

The highest emergy self-sufficiency ratio (ESR) values are for Zhaoqing and Jiangmen, both above 50%, followed by Zhongshan, Guangzhou, Foshan, Huizhou, Zhuhai, and Dongguan. Hong Kong and Macau have the lowest self-sufficiency rates, followed by Shenzhen, which is below 1%. The environmental loading rate (ELR) of the GBA city cluster is generally large because R is much lower than the sum of the emergy of nonrenewable resources and products and imports. The value for Hong Kong is as high as $1.17 \times 10^7$ and that for Macau is as high as $1.55 \times 10^6$, which are much higher than those of other cities and cannot be marked on one graph; the remaining cities in order of size are Shenzhen, Dongguan, Zhuhai, Guangzhou, Zhongshan, Foshan, Huizhou, Jiangmen, and Zhaoqing, with the lowest value of $5.51 \times 10^2$ being for Zhaoqing, indicating a relatively smaller environmental load, but relative to the national environmental load rate of 3.13 in 2016 [42], this is still too large. The ERR of the GBA city cluster is generally low, with Zhaoqing, Jiangmen, and Huizhou being relatively high and Hong Kong and Macau being the lowest

and unable to be marked on a single map. The low W values for Hong Kong and Macau indicate that the above two cities have high waste disposal rates and recyclable utilization rates, while Zhaoqing is as high as 10.1%, Foshan is 7.7%, Jiangmen is 7.3%, and Zhongshan is 6.0%, which are ranked high, indicating that the above four cities have great potential for waste disposal and the development of a circular economy. The EWI is generally high, with Macau reaching $4.49 \times 10^3$, followed by Shenzhen, Hong Kong, and Dongguan, and the lowest, Zhaoqing, is at $5.56 \times 10$. This indicates that the W in the GBA city cluster greatly exceeds the ERR, and the recyclable value of the waste needs to be explored vigorously. Among them, Hong Kong and Macau, with their special geographical location and small area, have limited potential to improve the emergy of the R value (Table 1, Figure 6).

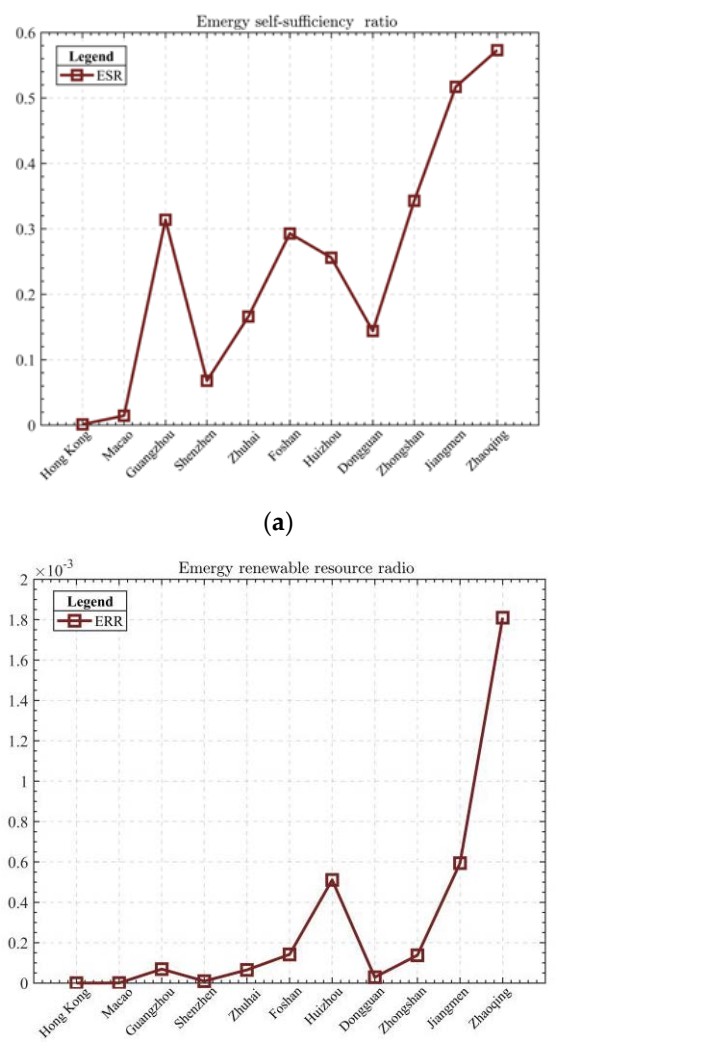

(a)

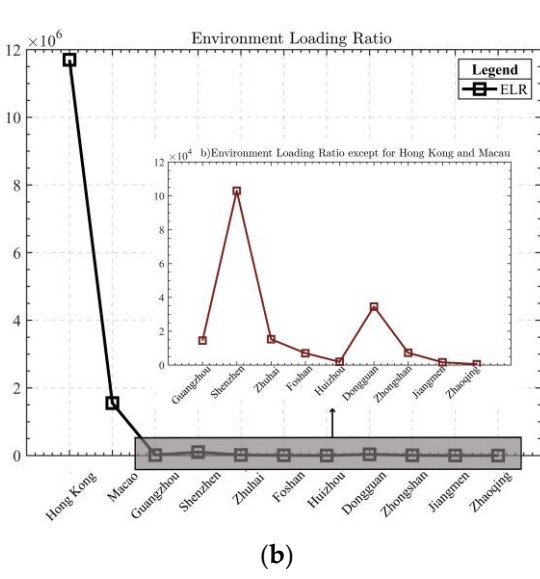

(b)

(c)

(d)

**Figure 6.** *Cont.*

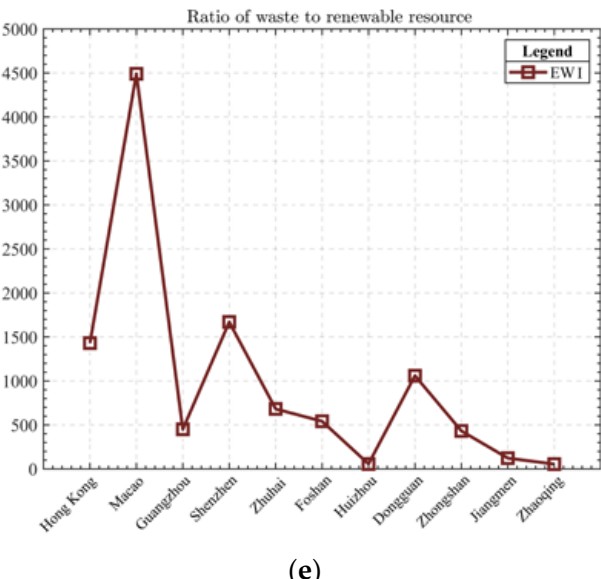

(**e**)

**Figure 6.** Natural subsystem emergy index (**a**) Comparison of Emergy self-sufficiency ratio (ESR) by cities in the GBA; (**b**) Comparison of Environment loading ratio (ELR) by cities in the GBA; (**c**) Comparison of Emergy renewable resource radio (ERR) by cities in the GBA; (**d**) Comparison of Emergy waste ratio (EWR) by cities in the GBA; (**e**) Comparison of Ratio of waste to renewable resource (EWI) by cities in the GBA.

### 3.5. Sustainable Development Analysis

The ESI, EISD, EUEHUI, and EUEHUI' are used to perform a comparative analysis of the sustainable development emergy of the system, which cannot be marked on the graph because the order of magnitude of the relevant indices of Zhaoqing is much larger than those for the other cities, but they are discussed together. The results show that all cities in the GBA have ESI values of <1 and are consumption-oriented [43]. Among them, the directions of the ESI and EISD are essentially the same, and the final ranking in order of quantity is Zhaoqing, Jiangmen, Huizhou, Foshan, Guangzhou, Zhongshan, Zhuhai, Dongguan, Macau, Shenzhen, and Hong Kong. In contrast, the directions of the EUEHI and EUEHI' are quite different, with the top three EUEHI' values in order of quantity being Zhaoqing, Macau, and Foshan, while the top three EUEHI' values in order of quantity are Zhaoqing, Jiangmen, and Huizhou (Table 1, Figure 7). In general, the ESI, EISD, EUEHI, and EUEHI' values all have very different directions, mainly because these indices only select individual combinations of indicators of urban eco-economic systems while not considering all indicators of the system, which is inevitably questionable. Li (2006) and Liu (2018) pointed out that at the national scale, the ELR is usually small due to the large amount of renewable resources available to them, while at the urban scale, the ELR is usually large due to the lack of resources, which often leads to a small quotient between the EYR and ELR (ESI) [16,21]. It is inappropriate to compare the ESI values at different scales. Table 1 shows that the ESI values of cities in the GBA are small and close to zero, making it difficult to compare them.

To evaluate the system sustainability more comprehensively, all indicators of the natural, social, and economic subsystems are analyzed with the help of the entropy method. First, all data for the three social–economic–natural subsystems in Table 1 are standardized, and the weights of each indicator are obtained using the entropy method, as shown in Table 2.

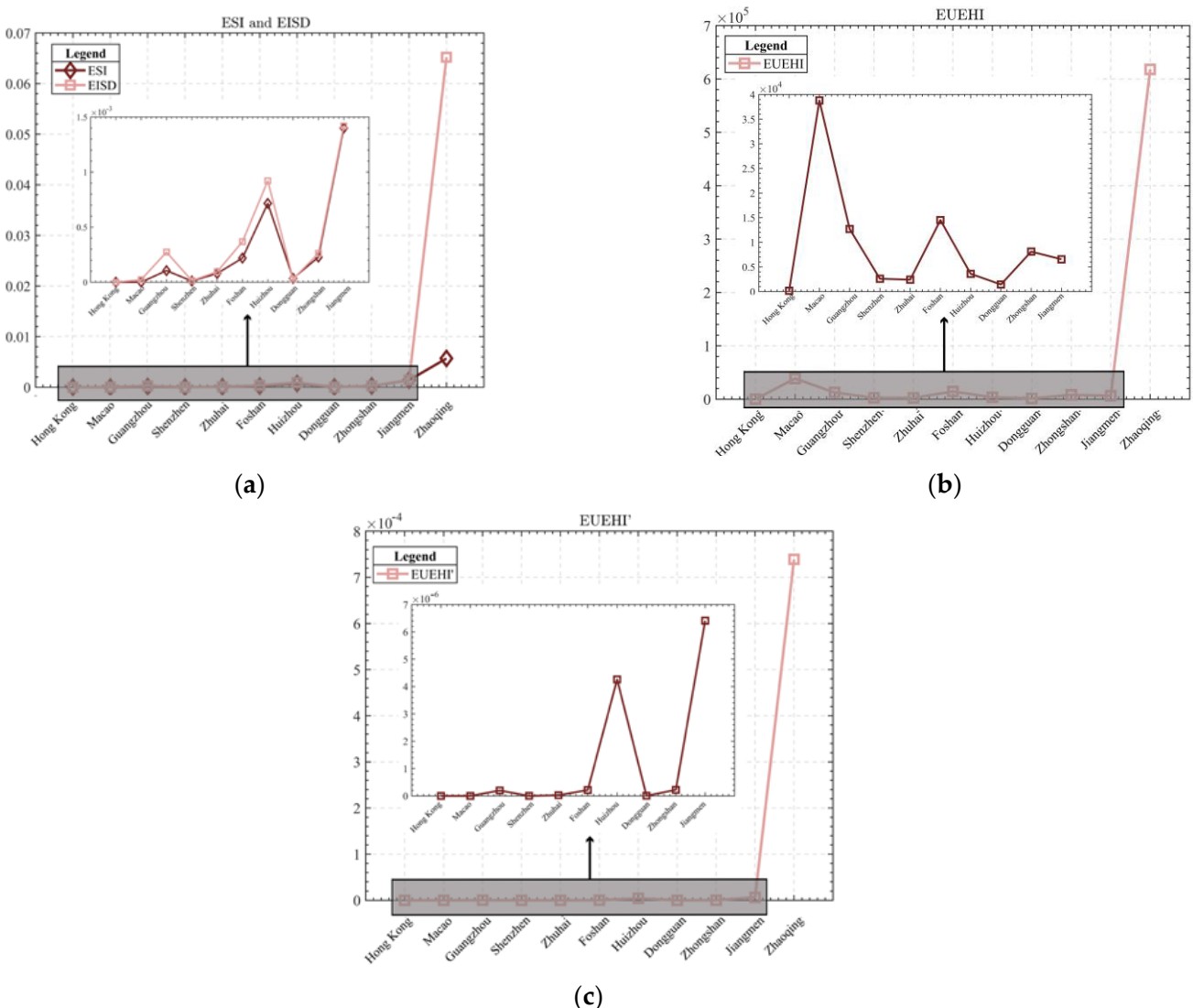

**Figure 7.** Emergy sustainability index and health emergy index of city eco-economic systems (**a**) Comparison of Emergy Sustainable index (ESI) and Emergy index for sustainable development(EISD) by cities in the GBA; (**b**) Comparison of Emergy-based urban ecosystem health index (EUEHI) by cities in the GBA; (**c**) Comparison of Modified emergy-based urban ecosystem health index (EUEHI') by cities in the GBA.

**Table 2.** Standardized values and weights of sustainable development capacity of the GBA in 2018.

| Social–Economic–Natural Subsystems | Index System | Hong Kong | Macau | Guangzhou | Shenzhen | Zhuhai | Foshan | Huizhou | Dongguan | Zhongshan | Jiangmen | Zhaoqing | Weight |
|---|---|---|---|---|---|---|---|---|---|---|---|---|---|
| Social Subsystem Indicators | ED | 0.000 | 0.871 | 1.000 | 0.990 | 1.000 | 1.000 | 1.000 | 1.000 | 1.000 | 1.000 | 1.000 | 0.040 |
| | EUC | 1.000 | 0.042 | 0.003 | 0.008 | 0.007 | 0.001 | 0.002 | 0.005 | 0.001 | 0.001 | 0.000 | 0.270 |
| | PCC | 0.170 | 0.000 | 9.160 | 11.200 | 1.460 | 4.800 | 3.230 | 6.590 | 1.880 | 2.220 | 1.000 | 0.060 |
| | EEC | 0.380 | 0.786 | 0.437 | 0.551 | 0.947 | 0.000 | 0.804 | 1.000 | 0.856 | 0.411 | 0.250 | 0.020 |
| Economic Subsystem Indicators | EPR | 0.000 | 0.049 | 0.405 | 0.191 | 0.297 | 0.515 | 0.601 | 0.398 | 1.000 | 0.813 | 0.990 | 0.040 |
| | EYR | 0.000 | 0.008 | 0.245 | 0.040 | 0.100 | 0.254 | 0.188 | 0.090 | 0.305 | 0.639 | 1.000 | 0.070 |
| | EMR | 0.000 | 0.811 | 0.947 | 0.886 | 0.881 | 0.975 | 0.906 | 0.854 | 0.960 | 0.935 | 1.000 | 0.040 |
| | EER | 0.000 | 1.000 | 0.057 | 0.020 | 0.017 | 0.032 | 0.018 | 0.005 | 0.016 | 0.012 | 0.340 | 0.170 |
| Natural Subsystem Indicators | ESR | 0.000 | 0.025 | 0.548 | 0.119 | 0.290 | 0.512 | 0.446 | 0.251 | 0.598 | 0.902 | 1.000 | 0.040 |
| | ELR | 0.000 | 0.867 | 0.999 | 0.991 | 0.999 | 0.999 | 1.000 | 0.997 | 0.999 | 1.000 | 1.000 | 0.040 |
| | EWR | 1.000 | 0.971 | 0.690 | 0.839 | 0.559 | 0.235 | 0.712 | 0.697 | 0.409 | 0.275 | 0.000 | 0.020 |
| | REE | 0.000 | 0.000 | 0.038 | 0.005 | 0.036 | 0.079 | 0.282 | 0.016 | 0.076 | 0.328 | 1.000 | 0.130 |
| | EWI | 0.690 | 0.000 | 0.911 | 0.635 | 0.859 | 0.890 | 1.000 | 0.774 | 0.915 | 0.985 | 1.000 | 0.050 |

The ecological and economic system indicators and sustainability indices of the GBA city cluster in 2018 were obtained after calculation via the entropy value method, as shown

in Table 3. From the sustainable development index, Hong Kong reaches the highest level at 0.262, followed by Zhaoqing and Macau, and then by Jiangmen, Huizhou, Guangzhou, Foshan, Zhongshan, Shenzhen, Dongguan, and other cities, while Zhuhai has the lowest level at only 0.019. However, there is a limitation to simply examining the sustainable development index, and it is also necessary to consider the coordination of the system, that is, to look at both the development of the system and whether the various subsystems of the system are coordinated.

**Table 3.** The eco-economic system indicators, sustainable development index, coordination degree, and coordination development degree of the GBA in 2018.

| Eco-Economic System and Coordinated Development | Hong Kong | Macau | Guangzhou | Shenzhen | Zhuhai | Foshan | Huizhou | Dongguan | Zhongshan | Jiangmen | Zhaoqing |
|---|---|---|---|---|---|---|---|---|---|---|---|
| Social Subsystems | 0.256 | 0.025 | 0.007 | 0.004 | 0.004 | 0.005 | 0.013 | 0.004 | 0.004 | 0.013 | 0.033 |
| Economic Subsystem | 0.000 | 0.092 | 0.016 | 0.006 | 0.007 | 0.010 | 0.010 | 0.008 | 0.012 | 0.017 | 0.055 |
| Natural Subsystems | 0.006 | 0.006 | 0.015 | 0.011 | 0.008 | 0.013 | 0.017 | 0.010 | 0.012 | 0.024 | 0.061 |
| Sustainable Development Index | 0.262 | 0.123 | 0.038 | 0.021 | 0.019 | 0.028 | 0.040 | 0.021 | 0.028 | 0.054 | 0.150 |
| Coordination Degree | 0.000 | 0.203 | 0.845 | 0.727 | 0.859 | 0.801 | 0.927 | 0.853 | 0.730 | 0.908 | 0.905 |
| Coordinated Development Degree | 0.000 | 0.158 | 0.179 | 0.124 | 0.127 | 0.150 | 0.192 | 0.135 | 0.143 | 0.222 | 0.369 |

According to Equation (6), we can calculate the coordination degree of the eco-economic system of each city in the GBA. The lowest is almost zero in Hong Kong, followed by Macau, both of which have low coordination degrees. Values for the remaining cities all range from 0.73 (Shenzhen and Zhongshan) to 0.93 (Huizhou), indicating a good degree of coordination. However, low levels of development of urban eco-economic systems may also cause high levels of coordination. Therefore, the sustainable and healthy development of a city should be the best state after coupling and coordinating between the sustainability index and coordination degree. According to Equations (7) and (8), we further calculate the coordinated development degree of each city as shown in Table 3, and the final ranking in order of quantity is Zhaoqing, Jiangmen, Huizhou, Guangzhou, Macau, Foshan, Zhongshan, Dongguan, Zhuhai, Shenzhen, and Hong Kong, which are different from the previous rankings of the ESI, EISD, EUEHUI, and EUEHUI′ (Table 3, Figures 7 and 8). The most typical representative of this case is Hong Kong, which has a high sustainable development index value and the lowest degree of coordination, with its social subsystem development being a standout, while the economic and natural subsystems are significantly underdeveloped, showing a clear incoherence within the system.

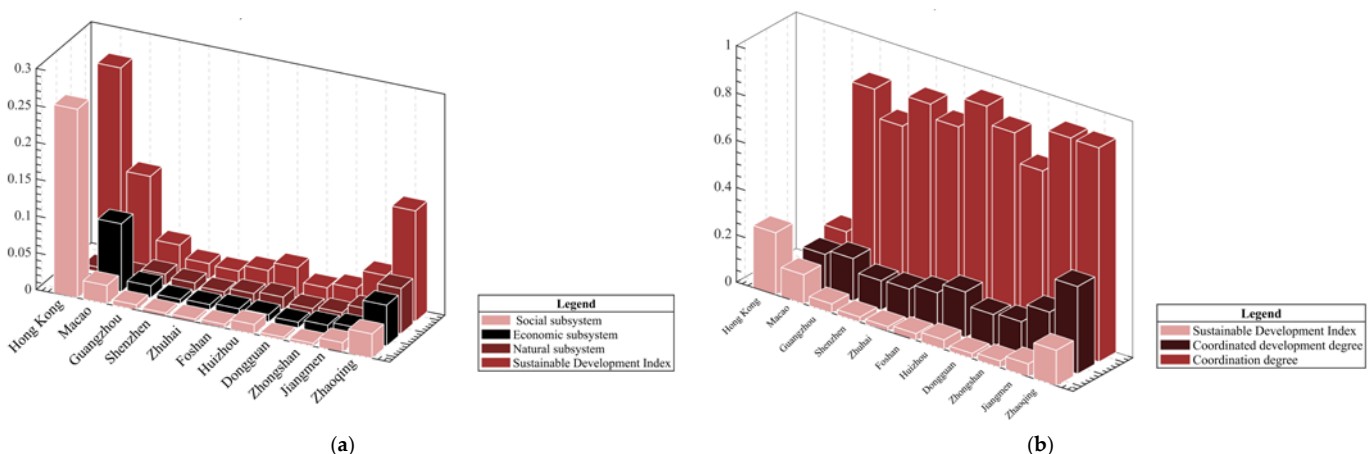

(**a**)　　　　　　　　　　　　　　　　　　　　　　　　(**b**)

**Figure 8.** Eco-economic system and coordinated development of the GBA (**a**) Comparison of Social subsystem, Economic subsystem, Natural subsystem and Sustainable Development Index by cities in the GBA; (**b**) Comparison of Sustainable Development Index, Coordination degree and Coordinated development degree by cities in the GBA.

## 4. Discussion

In recent decades, rapid urbanization and intensified global climate change have led to significant differences in the spatial distribution of environment and resources, which makes it difficult to accurately assess regional ecological sustainable development, especially in large urban agglomerations [9]. This study attempted to analyze the emergy flow, material flow, currency flow, population flow, waste logistics, and other types of flows inside and outside urban agglomerations (Figure 1), but the flow of information and culture flow in the urban eco-economic system lack in-depth analyses, and the information flow and culture flow should also be regarded as integral parts of the urban eco-economic system. The study and analysis of them can be more objective and accurate and could help to grasp the emergy flow track and exchange process of the ecological economic system. Facing the threat of an economic recession, achieving sustainable development and harnessing its economic and social benefits, while taking into account the quality of life and ecological environment, have become new and important topics to be explored. Many countries are focused on achieving the goals of sustainable growth and development, encompassing economic, social, and environmental dimensions [44]. The sustainable development of urban eco-economic system needs to be analyzed and studied from the perspective of diversification and internationalization [45], which will be the future research direction.

Many indices, such as the ESI, EISD, EUEHI, and EUEHI', have been used to evaluate the sustainability capacity of urban eco-economic systems. Based on the analysis in Figures 6 and 7, the ESI, EISD, EUEHI, and EUEHI' have very different trends of change compared with the indicators of sustainable development emergy obtained by the entropy value method. The common feature of the ESI and other types of sustainability indices for evaluating ecological and economic systems is that they are a combination of several indicators, which do not comprehensively take into account the data for natural, social, ecological, and economic subsystems and are inevitably biased. The sustainable development capability obtained by the entropy value method for the first time in this study shows the system's sustainable development capability more comprehensively and systematically because all indicators of the three subsystems, namely social, economic, and natural, are comprehensively introduced for evaluation. Moreover, combining the development degree with the coordination degree can take both into account to obtain the coordinated development degree, which can finally express the sustainable development capability of both the coordination and development. Based on the results of the calculations, the cities can be divided into three categories, allowing for an analysis of the different urban emergy -value characteristics within the urban agglomerations (Table 4).

**Table 4.** List of city emergy characteristics in the GBA.

| City Name | Category | Degree of Coherent Development | Nature of the City | Emergy Characteristics |
|---|---|---|---|---|
| Hong Kong | 1 | 0.000 | International financial, shipping, and trade center and international aviation hub | Hong Kong has the best import and export emergy levels, ESR, and high sustainable development capacity in the GBA, but the excessive total consumption emergy leads to the highest ED, EUC, EPR, and ELR values among the GBA city groups, while the EYR, EER, and coordination level are the lowest. The overreliance on import emergy leads to Hong Kong's economic growth lacking sustainable and solid support, and its development space faces bottleneck constraints. |

**Table 4.** *Cont.*

| City Name | Category | Degree of Coherent Development | Nature of the City | Emergy Characteristics |
|---|---|---|---|---|
| Shenzhen | 1 | 0.124 | A modern city with high technology, finance, and logistics as pillars | Shenzhen's EMR and ERR are high, which can make better use of renewable resources. The high emergy from waste and population-carrying capacity and the larger share of imported emergy than exported emergy indicate that the urban ecological environment is under great pressure, and the population far exceeds the urban land area and population-carrying capacity. |
| Zhuhai | 1 | 0.127 | The gateway hub city of the GBA | Zhuhai's electricity emergy use ratio, EEC, and EMR are in the top four, while the ESR is in the second-to-last position, which proves that Zhuhai is a city with a higher dependence on nonrenewable resources such as electricity, relies more on imported emergy, and needs to consume more emergy per unit of production. |
| Dongguan | 1 | 0.135 | A city characterized by foreign trade with manufacturing as its main pillar | Dongguan's nonrenewable resources and product emergy, ED, and total consumption emergy are all high. The city's resource allocation is fragmented and consumed, and the product output efficiency is low; the land development intensity is as high as 49.4%, much higher than the international warning line of 30% [34]. |
| Zhongshan | 2 | 0.143 | Important hub city of the coastal economic belt of the GBA city cluster | The overall emergy characteristics of Zhongshan are close to those of Foshan, but the overall emergy consumption is lower, indicating that the quality and standard of living of its residents are slightly lower than for Foshan, and the emergy is more derived from the self-generated natural ecosystem, which reduces the dependence on imported emergy to a certain extent, but the environmental pressure and population-carrying capacity are still larger. |
| Foshan | 2 | 0.150 | The GBA's western integrated hub city | The emergy characteristics of Foshan are very close to those of Guangzhou, with high land development intensity and obvious fragmentation and slightly lower economic development dynamics than the other cities [37], but the consumption of emergy and environmental pressure are less than Guangzhou's, and the recycling rate of waste is higher than Guangzhou's. |
| Macau | 2 | 0.158 | World tourism and leisure center | Macau has the second highest ED, EMR, EEC, ELR, EWR, and ERR values after Hong Kong but has the highest ESR and EWI values and the lowest waste emergy in the GBA city cluster. Macau needs to maintain the city by reducing the total emergy consumption and the recycling of waste emergy under the situation of land space resource scarcity and high intensity levels of land development, integration, and sustainable development. |

**Table 4.** *Cont.*

| City Name | Category | Degree of Coherent Development | Nature of the City | Emergy Characteristics |
|---|---|---|---|---|
| Guangzhou | 2 | 0.179 | The capital of Guangdong Province, a cosmopolitan city integrating commerce, science, education, and culture | Guangzhou's total emergy consumption is only lower than that of Hong Kong and Shenzhen, and its waste emergy and population-carrying capacity remain high. Guangzhou's emergy consumption is high, while its waste reuse rate is too low, and its population base is large, resulting in excessive population-carrying pressure. |
| Huizhou | 3 | 0.192 | An important node city in the Guangdong–Hong Kong–Macau Greater Bay Area | Huizhou's emergy flow indicators are slightly higher than Jiangmen, except for N and W. The social subsystem indicators are higher than Jiangmen, while the economic subsystem indicators are lower than Jiangmen except for EMR, and the natural subsystem indicators are lower than Jiangmen except for the environmental load rate. This indicates that Huizhou's ecological and economic benefits are slightly lower than those of Jiangmen, and it is under greater environmental pressure. |
| Jiangmen | 3 | 0.222 | Hub gateway city of the western wing of the GBA | Jiangmen's emergy characteristics are closer to those of Zhaoqing, and the emergy flow indicators are higher than those of Zhaoqing, except for R and the export emergy, which are lower. The social subsystem indicators are higher than those of Zhaoqing, while the economic subsystem indicators are lower and the natural subsystem indicators are higher except for ESR. This indicates that Jiangmen's overall emergy consumption is higher than that of Zhaoqing, as well as its resident living standards. The overall emergy consumption of Jiangmen is higher than that of Zhaoqing, the quality and standard of living of the residents are higher than that of Zhaoqing, and there is a certain amount of economic development emergy, but the ecological and economic benefits are lower than those of Zhaoqing. |
| Zhaoqing | 3 | 0.369 | An important node city in the GBA | Zhaoqing's renewable resources and product emergy are the highest in the GBA city cluster, while the incoming and export emergy levels, total consumption emergy, ED, and EUC are the lowest, indicating that Zhaoqing's emergy consumption structure is relatively more optimized, but the resident's quality of life and living standards are the lowest among the 11 cities in the whole GBA; however, the EYR is the highest and the ELR is the lowest, indicating that the city's ecological economy is high, while the pressure on the environment is relatively low. |

Many previous studies have not given much consideration to the confusion of emergy conversion rates due to emergy baselines, but the continuous updating of emergy baselines has strengthened the basis for accounting for nonvalue quantities of ecosystem services, and there is a need to clarify the emergy baseline values used in emergy accounting [46]. In the emergy handbook, Folio #1, Odum et al. concluded that the baseline value of the emergy conversion rate should be increased from $9.44 \times 10^{24}$ sej/a in 1996 to $15.83 \times 10^{24}$ sej/a [13]. Brown et al. recalculated the emergy baseline to

$15.20 \times 10^{24}$ sej/a [47]. These values are different and should be clarified before performing a study.

## 5. Conclusions

### 5.1. Common Problems of the GBA City Cluster

The common characteristics of the GBA city group are that the coordinated development degree is low and that the cities are resource-consuming; the ERR is very low, which indicates that there is great potential for developing renewable emergy and resources, while the high EWI indicates that the recycling rate of waste is low, and there is a large amount of waste from nonrenewable resources. The tension between humans and land in cities other than Zhaoqing, Jiangmen, and Huizhou indicates that the effective use of land resources has become a bottleneck for sustainable development. The emergy structure of the whole GBA city cluster is extremely unreasonable. The flow of emergy among city clusters should be promoted to form a graded and differentiated industrial development layout with mutual collaboration and a reasonable division of labor. The development momentum, innovation capacity, and transformation of scientific and technological achievements within city clusters should be further enhanced. With the deepening of the supply-side structural reform, new industries and new models should be further developed based on the upgrading and transformation of traditional industries [23], which will promote the transformation of the GBA city cluster from resource-consuming cities to world-class innovative industrial city clusters. Efficient and fast transportation networks between cities should be formed to improve the material and population flows between city clusters. As for the Pearl River, Xijiang, Dongjiang, Beijing, and Hanjiang, the total amount of pollutants entering the sea from the river network of the Pearl River Delta should be controlled, the comprehensive environmental improvement of black and smelly water bodies along rivers should be deepened, and the level of resource utilization of solid waste should be improved.

### 5.2. Classification of City Clusters from an Emergy Perspective

Based on the results of the emergy development coordination degree calculation, combined with the results of emergy flow and the social–economic–natural subsystem index data calculation, the GBA city cluster can be divided into three categories.

(1) Category 1 cities have coordinated development degrees between 0.0 and 0.135, mainly including four cities, Hong Kong, Shenzhen, Dongguan, and Zhuhai, whose main emergy characteristics include having the highest ED in the GBA city cluster, indicating that the economic development of this category of cities is faster, but the contradiction between people and land is prominent, while the land area is a serious constraint to sustainable development; the EUC values are higher, with the EUC value of Hong Kong being the highest, with Dongguan, Zhuhai, and Shenzhen being just below Macau, which indicates that the residents of these cities have higher living standards and quality. The lower EYR indicates that their eco-economic efficiency is lower in the GBA; the lowest ESR and the highest environmental load rate indicate that the dependence on external emergy is extremely high, and the system is under great environmental pressure.

(2) Category 2 cities have coordinated development degrees ranging between 0.143 and 0.179, mainly including four cities: Guangzhou, Macau, Foshan, and Zhongshan. Their main emergy characteristics are the emergy flow, social subsystem, economic subsystem, and natural subsystem, which are all between category 1 and category 3 cities and are in the middle of the whole GBA.

(3) Category 3 cities have coordinated development degrees ranging between 0.192 and 0.369, mainly including Zhaoqing, Jiangmen, and Huizhou, which are cities with low ED and EUC levels in the GBA city cluster. This indicates that the land area still has some potential, but the residents' living standards and quality of life are relatively low. The ESR is high, with Zhaoqing and Jiangmen even exceeding 50%, indicating a high natural environmental support capacity; however, the ELR (102–103 orders of

magnitude) is still much higher than the national average ($10°$ orders of magnitude, 2016) [42], indicating that the system's environmental pressure remains high.

Overall, in terms of the economic development and innovation development levels, the cities are ranked as category 1 > category 2 > category 3. In terms of the ecological and environmental conditions and blue–green space protection, the cities are ranked as category 1 < category 2 < category 3. Category 2 is between category 1 and category 3 in both instances.

*5.3. Empirical Evidence Shows That Emergy Analysis Can Effectively Evaluate the Sustainability of Urban Cluster Eco-Economic Systems*

As an ecological accounting method based on eco-thermodynamics with a holistic perspective, no other accounting framework can achieve such a uniform and integrated quantification of components and flows in a system. Scholars have shown in the past that emergy analyses can be effectively used to study regional eco-economic systems at the national, provincial, and city levels, and the results of this study perfectly demonstrate that emergy analyses are equally fruitful and practical for evaluating the sustainability of the eco-economic systems of the Greater Bay Area city cluster. This study highlights the characteristics of the socioeconomic–natural subsystems of each city in the city cluster, provides a horizontal comparison of the problems of each city for a comprehensive analysis, and proposes optimized countermeasures.

**Author Contributions:** H.L.: Conceptualization, methodology, investigation, writing—original draft, writing—review and editing, and supervision. X.H.: Conceptualization and methodology. Q.X.: Conceptualization, methodology, and investigation. S.W. and W.G.: Investigation. Y.L.: Investigation. Y.H.: Writing—review and editing. J.W.: Supervision. All authors have read and agreed to the published version of the manuscript.

**Funding:** This research was funded by National Natural Science Foundation of China (52078222) and the Key Scientific Research Project of Colleges and Universities of Guangdong Education Department in 2020 (2020ZDZX1033).

**Acknowledgments:** We would like to thank the National Natural Science Foundation of China (52078222) and the Key Scientific Research Project of Colleges and Universities of Guangdong Education Department in 2020 (2020ZDZX1033) for its support. We also thank the anonymous reviewers for their helpful and valuable comments and suggestions to improve the quality of the study.

**Conflicts of Interest:** The authors declare no conflict of interest.

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
