# Peer review of "A New Approach to Evaluate the Sustainability of Ecological and Economic Systems in Megacity Clusters: A Case Study of the Guangdong–Hong Kong–Macau Bay Area"

_sustainability, doi:10.3390/su15075881_

Round 1

Reviewer 1 Report

Comments:

The authors proposed a new approach for evaluating the sustainability of ecological and economic systems in megacity clusters. The structure of the paper is good but some comments are listed to revise the paper.

1. Please underscore the scientific value-added of your paper in your abstract and introduction.

2. Some chapters describe too much content, it is recommended to briefly explain. (For example abstract and results and discussion)

3. The titles of Chapter 3 "Results and discussion" and Chapter 4 "Discussion" are not very good, both have "discussion". It is suggested that the author can revise the suitable title.

4. In the Introduction and Literature review, the author conducts detailed literature discussions on methods and others..., but it would be better if the literature can be added in the last five years.

5. Table 1 in Chapter 2 is very important, and it is recommended that the authors elaborate more on the content, and add arguments and contributions.

6. It is very important for the author to choose the research method, which can increase the literature citation and strengthen the rationality.

7. Please check the parameter descriptions of all formulas in this article to increase the readability.

8. To be legible, the whole text must be completely edited with the help of a native English editor to polish your writing to prevent redundancies, grammatical errors, and punctuation problems.

In a summary, I recommend the paper be accepted for publication after the above changes.

Author Response

Point 1: Please underscore the scientific value-added of your paper in your abstract and introduction.

Response 1: We further clarify the purpose and significance of this study in the abstract, highlighting the importance of this study for the coordinated development of cities and cities in urban agglomerations for stable and sustainable regional development, and inserting relevant references from the last five years in the introduction to support the scientific significance of this study.(For details see: summary and lines77 , 103 of the introduction on page 3

Point 2: Some chapters describe too much content, it is recommended to briefly explain. (For example abstract and results and discussion).

Response 2: We have mainly simplified the lengthy sections of the text by presenting the conclusions in a table.(For details see: Table 4 on page 19

Point 3: The titles of Chapter 3 "Results and discussion" and Chapter 4 "Discussion" are not very good, both have "discussion". It is suggested that the author can revise the suitable title.

Response 3: We have changed Chapter 3 "Results and Discussion" to "Results and Analysis", Chapter 4 to "4. Discussion" and Chapter 5 has been changed to "5. Conclusions" for clarity and completeness.For details see: line 313 on page 10, line 500 on page 18, line 549 on page 21

Point 4: In the Introduction and Literature review, the author conducts detailed literature discussions on methods and others..., but it would be better if the literature can be added in the last five years.

Response 4: We have added references from the last five years in the introductory section of the article in terms of research progress and application of methods.(For details see: line 77 , 103 of the introduction on page 3, line 504, 515, 517 on page 18

Point 5: Table 1 in Chapter 2 is very important, and it is recommended that the authors elaborate more on the content, and add arguments and contributions?

Response 5: We have made further adjustments to all tables and tables in the article, including the harmonisation of data formatting and the addition of references to some data descriptions .(For details see: line 240 on page 7line 471、481 on page 17, Adding literature citations line 241、242 on page 8

Point 6: It is very important for the author to choose the research method, which can increase the literature citation and strengthen the rationality.

Response 6: To clarify the importance of the research methodology as well as its soundness, we have added references to the last five years in the introduction to the article.(For details see: line 77,103 on page 2

Point 7: Please check the parameter descriptions of all formulas in this article to increase the readability.

Response 7: We further explain the entropy weighting formula (4) and the systematic evaluation formula (5) in the text to clarify the main role of the individual institute in the evaluation method in order to enhance the readability of the article.(For details see: line 273-280 on page 9

Point 8: To be legible, the whole text must be completely edited with the help of a native English editor to polish your writing to prevent redundancies, grammatical errors, and punctuation problem.

Response 8: After revising the review comments, we submitted a language edit to the relevant department of the journal.

Reviewer 2 Report

This paper mainly focuses on the evaluation of ecological and economic systems in megacity clusters, but the method proposed is not innovative, only uses more indicators. Also, the structure of the paper is chaotic. I think the author did not conduct a strict format review before submitting the manuscript, and the presentation of the charts is not clear, making it very difficult to read. The specific suggestions are as follows:

1.      English needs careful inspection and polishing.

2.      For Figures 1 and 5, the font size is too small to distinguish.

3.      The content in Table 3 is reserved to the last three decimal places, please unify.

4.      I don't understand why the reference starts from [22], I have never seen this kind of citation format.

5.      Line 83, Is the format of this reference correct?

6.      Line 83-92, this part should belong to the methods section.

7.      Please clarify the research objectives of this paper.

8.      Data source, using this kind of list-style introduction is not a good way, a simple introduction is enough.

9.      The information expressed in Figure 1 is too confusing, because the font is too small, a lot of information cannot be read clearly, which makes it difficult for people to understand what the overall framework of the method is. It is recommended to draw a new clear method framework.

10.   For the evaluate method proposed by the author, only more indicators are considered. I don’t think there is any obvious innovation in this, because many indicators are repeated or have strong correlations. The result of considering more indicators calculation is not necessarily better.

11.   Funding and Acknowledges are exactly the same, there is no need to write Acknowledges part.

Author Response

Point 1: English needs careful inspection and polishing.

Response 1: After revising the review comments, we submitted the language touch-ups to the journal.

Point 2: For Figures 1 and 5, the font size is too small to distinguish.

Response 2: All images and text in the article have been changed to ensure that the readers can see them clearly.For details see: on page 6, 14

Point 3: The content in Table 3 is reserved to the last three decimal places, please unify.

Response 3: We have aligned the data in Table 2 and Table 3 in the text to retain three decimal places.For details see: on page 17

Point 4: I don't understand why the reference starts from [22], I have never seen this kind of citation format.

Response 4: We have further sorted through the references and added some references based on the template provided in the journal pair (For details see: 23 pages, references section).

Point 5: Line 83, Is the format of this reference correct?

Response 5We modified the article according to the article citation format.(For details see: line 89 on page 2)

Point 6: Line 83-92, this part should belong to the methods section.

Response 6: Lines 83-92 of the original text focus on research advances and frontiers in energy value analysis methods, and are therefore not integrated with the methods section at this time.For details see: line 89-98 on page 2

Point 7: Please clarify the research objectives of this paper.

Response 7:In the "Abstract" and "Introduction", we further clarify the objective of the study, which is to further optimise the scientific validity of entropy analysis methods for the evaluation of complex urban systems.For details see: line 12-15 on page 1, line 51-54 on page 2

Point 8: Data source, using this kind of list-style introduction is not a good way, a simple introduction is enough.

Response 8: We have simplified the parts of the data source that are duplicated.(For details see: line 173-175 on page 5

Point 9: The information expressed in Figure 1 is too confusing, because the font is too small, a lot of information cannot be read clearly, which makes it difficult for people to understand what the overall framework of the method is. It is recommended to draw a new clear method framework.

Response 9: We add the technical lines of the article in article 2.1.For details see: on page 4

Point 10: For the evaluate method proposed by the author, only more indicators are considered. I don’t think there is any obvious innovation in this, because many indicators are repeated or have strong correlations. The result of considering more indicators calculation is not necessarily better.

Response 10: Thank you very much for pointing out the problem.An emergy analysis is used to assess the sustainability of urban agglomerations' eco-economic systems, which are generally measured by emergy–value sustainability indicators using a combination of several system indicators. However, this assessment approach is not applicable to economically developed high-density urban agglomerations.The results of the coupling analysis between the indicator systems are very much in line with the "social-economic-ecological" problems that cities are facing today. Therefore, compared to the ordinary entropy method, the comprehensive selection of factors of different indicator systems and the coordinated coupling analysis can better guide the sustainable development of megacities.(For details see:  Abstract section on page 1, Discussion section on page 18, Conclusions section on page 21

Point 11: Funding and Acknowledges are exactly the same, there is no need to write Acknowledges part.

Response 11: The acknowledgement section has been removed by us.(For details see: on page 23)

Reviewer 3 Report

Several questions should be carefully addressed. Please see below for the comments, which may be helpful for improving the quality of the manuscript.

Abstract

The word count is excessive and the main findings of the article could be summarised in shorter terms.

Introduction

1. Please check carefully the format of the bibliographic citations in the article, some of them have missing spaces between the citation number and the article.

2. Line 115: "GBA" should be explained in this place.

Materials and Methods

1. Line 129 and 145: What does the underlining mean here?

2. Line 160: All secondary headings in the article are not formatted to meet journal requirements, please read the journal requirements carefully.

3. Line 161-171: The authors could have summarised the data in more concise terms.

4. Line 172-230: The title of this section is "Materials and Methods", but the authors have analysed the results in this section, which should be placed under "3. Results and discussion".

5. Figure 1: The authors should double check the font formatting of all figure titles and table captions in the text; the title of the figure in the journal should be "Figure" and is missing punctuation; the figure is not very clear, especially in the right-hand legend; the formatting of the city names in the figure does not correspond to the text.

6. Table 1: The authors need to re-layout the contents of the table as it seems to me that the words in this table are confusing.

7. Line 230: I think these indicators need to be explained in the article, rather than the reader searching for other literature; the spacing between the "Note" sentence and the explanation of the four indicators above is too large, and both sentences should be "left-justified".

8. Line 249, 260 and 267: You can use "2.4.1", "2.4.2" ......, instead of "(1)", "(2)" .......

9. Equation: All equations in the text are incorrectly formatted and the serial numbers of the formulas that follow are not aligned. I recommend using a formula editor to make the changes.

10. Line 294 and 302: There should be punctuation in this place.

Results

1. Line 303: The title format does not match the requirements of the journal.

2. Figure 2: Four diagrams have different dimensions; there is too much white space below diagram (a); there is no need to mark (a) (b) (c) (d) above the diagram when it is already marked below; (a) (b) and (c) (d) have different looking font formats.

3. Figure 3: Same as Figure 2.

4. Figure 4: Two diagrams are of different dimensions and (a) the label above the diagram is c)? Other requirements are the same as in Figure 1.

5. Figure 5 and Figure 6: The problem is the same as in Figure 1, so please revise it carefully.

6. Table 2 and Table 3: It seems to me that the content of this table is not formatted in the correct font, please adjust the content of the table to make it look better.

7. Figure 7: (a) (b) is labelled below, and a) b) is also labelled in the diagram, which is repeated, while the font size of the labelling below is different from that in the previous diagram.

Discussion

1. There is a discussion in the third part and a discussion in the fourth part. There is some confusion between the two parts, so please rearrange the layout of the article.

2. Line 513-514: There is a space between "10" and "24" in these data, but "1024" in line 223 does not have this space, is this a formatting error or does it mean something else?

Conclusions

1. Part of the conclusion should be shown in the discussion and the conclusion uses concise words to summarise the main findings of the article.

2. Line 585-586: The label "(1)" has been used earlier and should be followed by other numbers as a way of distinguishing the different classifications.

For these reasons, my final recommendation is the rejection of the manuscript.

Author Response

Abstract

Point 1:  The word count is excessive and the main findings of the article could be summarised in shorter terms. 

Response 1: To address this issue, we first provided a further high level overview of the article's subject matter, as well as linguistic editing of the English manuscript.

Introduction

Point 1: Please check carefully the format of the bibliographic citations in the article, some of them have missing spaces between the citation number and the article.

Response 1: We have further reviewed the format of all citations in the article.For details see: 23 pages, references section

Point 2: Line 115: "GBA" should be explained in this place.

Response 2: We have explained line 115: "GBA".(For details see: line 122,123 on page 3)

Materials and Methods

Point 1: Line 129 and 145: What does the underlining mean here?

Response 1: There was an editing error, we have removed the underlining.(For details see: line 139 on page 4 , line 155 on page 5 )

Point 2: Line 160: All secondary headings in the article are not formatted to meet journal requirements, please read the journal requirements carefully.

Response 2: We have made initial revisions to the format of the article by downloading the standard format and editing the language.(For details see: line 172 on page 4)

Point 3:  Line 161-171: The authors could have summarised the data in more concise terms.

Response 3: We have used more concise terminology for this section, and have also improved the language of the text to ensure quality reading for the reader.(for details see: line 173-175 on page 5

Point 4: Line 172-230: The title of this section is "Materials and Methods", but the authors have analysed the results in this section, which should be placed under "3. Results and discussion".

Response 4: Line 172-230 focuses on the process of energy flow and circulation in the various subsystems of the Guangdong-Hong Kong-Macao Bay, which we have summarised to form a general level system, and we have also interpreted the indicators for the selected areas.for details see: line 215-233 on page 6

Point 5: Figure 1: The authors should double check the font formatting of all figure titles and table captions in the text; the title of the figure in the journal should be "Figure" and is missing punctuation; the figure is not very clear, especially in the right-hand legend; the formatting of the city names in the figure does not correspond to the text.

Response 5: We have corrected the text formatting of the city in image 1, enlarged the image text characters, and enlarged the legend, etc.For details see: line 215-233 on page 6

Point 6: Table 1: The authors need to re-layout the contents of the table as it seems to me that the words in this table are confusing.

Response 6: In response to this question, we have included in the text "2.4. Accounting for the emergy of the GBA" a uniform description of the indicators covered in the article.(For details see: line 240 on page 7

Point 7: Line 230: I think these indicators need to be explained in the article, rather than the reader searching for other literature; the spacing between the "Note" sentence and the explanation of the four indicators above is too large, and both sentences should be "left-justified".

Response 7: We have modified the data in Table 1 and revised the marker annotations, and we have provided a general description of the significance of the overall indicator selection for each subsystem.For details see: line 224, 226, 230 on page 6, line 240 on page 7

Point 8: Line 249, 260 and 267: You can use "2.4.1", "2.4.2" ......, instead of "(1)", "(2)" ........

Response 8: We use "2.5.1", "2.5.2" instead of "(1)", "(2)" according to the format of the article and the comments .(For details see: line 261 on page 8, line 270, 278 on page 9

Point 9: Equation: All equations in the text are incorrectly formatted and the serial numbers of the formulas that follow are not aligned. I recommend using a formula editor to make the changes.

 Response 9: We re-edited a bunch of article formulas according to the article formatting specifications.(For details see: line 265 , 266, 276 , 279 ,296 , 303, 305 on page 9)

Point 10:  Line 294 and 302: There should be punctuation in this place.

 Response 10: Punctuation has been added.(For details see: line 296 on page 9, line 312 on page 10)

Results

Point 1: Line 303: The title format does not match the requirements of the journal.

 Response 1: We have restructured '3. Results and discussion' and revised the format with reference to the formatting standards.(For details see: line 313 on page 10)

Point 2: Figure 2: Four diagrams have different dimensions; there is too much white space below diagram (a); there is no need to mark (a) (b) (c) (d) above the diagram when it is already marked below; (a) (b) and (c) (d) have different looking font formats.

 Response 2:We have amended Figure 2 to agree the formatting and a more logical layout(For details see: line 357 on page 11)

Point 3: Figure 3: Same as Figure 2.)

 Response 3: We have modified Figure 3 in the same way as the processing of Figure 2.(For details see: line 384 on page 12)

Point 4: Figure 4: Two diagrams are of different dimensions and (a) the label above the diagram is c)? Other requirements are the same as in Figure 1.

Response 4:We have corrected Figure 4(a) by deleting the labels placed on the icons and enlarging the figure chapter appropriately in accordance with the treatment in Figure 1.For details see: line 419 on page 14

Point 5: Figure 5 and Figure 6: The problem is the same as in Figure 1, so please revise it carefully.

 Response 5: We have modified Figures 5 and 6 as required.(For details see: line 443 on page 15,line 445 on page 16)

Point 6: Table 2 and Table 3: It seems to me that the content of this table is not formatted in the correct font, please adjust the content of the table to make it look better.

 Response 6: Based on expert advice and typographical specifications, we have refined Table 2, added one column to Table 2 to more clearly indicate the table data in order to facilitate the reader's reading of the article's intent, and adjusted Table 3; the data in Tables 2 and 3 have been revised in a uniform manner.(For details see: line 471, 481 on page 17)

Point 7: Figure 7: (a) (b) is labelled below, and a) b) is also labelled in the diagram, which is repeated, while the font size of the labelling below is different from that in the previous diagram.

Response 7: In accordance with the typographic specifications, we have adjusted Figure 7 accordingly by removing the "a), b)" from the top of the image and enlarging it appropriately.(For details see: line 499 on page 18)

Discussion

Point 1: There is a discussion in the third part and a discussion in the fourth part. There is some confusion between the two parts, so please rearrange the layout of the article.

Response 1: In response to this problem, we have rearranged the layout of the article for the  fourth and fifth parts.(For details see: page 21)

Point 2:  Line 513-514: There is a space between "10" and "24" in these data, but "1024" in line 223 does not have this space, is this a formatting error or does it mean something else?

Response 2: This is due to a formatting error when converting Word to other question editing software, originally "1022, 1023, 1024, 1025,。。。。。。"(For details see: line 539-548 on page 21)

Conclusions

Point 1: Part of the conclusion should be shown in the discussion and the conclusion uses concise words to summarise the main findings of the article.

Response 1:In response to this problem, we have adapted the third, fourth and fifth parts of the article and simplified the recommendations.For details see: Conclusions section on page 11

Point 2: Line 585-586: The label "(1)" has been used earlier and should be followed by other numbers as a way of distinguishing the different classifications.

 Response 2: We have restructured this section to organize the juxtapositions into tables for easy comparison of individual cities, so "(1)" has been removed.(For details see: Discussion section on page 18-19)

Round 2

Reviewer 2 Report

I think the author has made careful revisions, and the problems I raised have been well resolved. The current version can be accepted.

Reviewer 3 Report

Thanks for you response. The revision meets the requirements and is recommended for acceptance.